

# Seasonal variation of fine- and coarse-mode nitrates and related aerosols over East Asia: Synergetic observations and chemical transport model analysis

Itsushi Uno[1], Kazuo Osada[2], Keiya Yumimoto[1], Zhe Wang[1,3], Syuichi Itahashi[4],
Xiaole Pan[3], Yukari Hara[1], Yugo Kanaya[5], Shigekazu Yamamoto[6], Thomas Duncan Fiarlie[7]

[1] Research Institute for Applied Mechanics, Kyushu University, Kasuga Park 6-1, Fukuoka 816-8580, Japan
[2] Nagoya University, Graduate School of Environ. Studies, Furo-cho, Chikusa-ku, Nagoya, 464-8601, Japan
[3] Institute of Atmospheric Physics, CAS, Beijing, China
[4] Central Research Institute of Electric Power Industry, Abiko, Chiba, 270-1194, Japan
[5] Japan Agency for Marine-Earth Science and Technology, 3173-25 Showa-machi, Kanazawa-ku, Yokohama, 236-0001,
Japan
[6] Fukuoka Institute of Health and Environmental Sciences, Mukaizano 39, Dazaifu, Fukuoka 818-0135, Japan
[7] NASA, Langley Research Center, Hampton, VA 23681-0001, U.S.A.

*Correspondence to*: Itsushi Uno (uno@riam.kyushu-u.ac)

**Abstract.** We analyzed long-term fine- and coarse-mode nitrate and related aerosols ($SO_4^{2-}$, $NO_3^-$, $NH_4^+$, $Na^+$, $Ca^{2+}$) synergetic observations at Fukuoka (33.52°N, 130.47°E) from August 2014 to October 2015. A Goddard Earth Observing System chemical transport model (GEOS-Chem) including dust and sea-salt acid uptake processes was used to assess the observed seasonal variation, and the impact of long-range transport (LRT) from the Asian continent. For fine aerosols
($fSO_4^{2-}$, $fNO_3^-$, and $fNH_4^+$), numerical results explained the seasonal changes, and a sensitivity analysis excluding Japanese domestic emissions clarified the LRT fraction at Fukuoka (85% for $fSO_4^{2-}$, 47% for $fNO_3^-$, 73% for $fNH_4^+$). Observational data for $HNO_3$, $fNO_3^-$, and coarse $NO_3^-$ ($cNO_3^-$) confirmed that $cNO_3^-$ made up the largest proportion of total nitrate (defined as the sum of $fNO_3^-$, $cNO_3^-$ and $HNO_3$), constituting 40−55% of total nitrate during the winter, while $HNO_3$ gas constituted approximately 40% of total nitrate in summer, and $fNO_3^-$ peaked during the winter. A numerical model reproduced the
seasonal variations in $fNO_3^-$. For $cNO_3^-$, large-scale dust-nitrate outflow from China to Fukuoka was confirmed during all dust events that occurred between January and June. Modeled $cNO_3^-$ was in good agreement with observations between July and November (mainly coming from sea salt-$NO_3^-$). However during the winter, the model underestimated $cNO_3^-$ levels compared to the observed levels. The reason for this underestimation was examined statistically using multiple regression analysis (MRA). We used $cNa^+$, nss-$cCa^{2+}$, and $cNH_4^+$ as independent variables to describe the observed $cNO_3^-$ levels; these
variables were considered representative of sea salt-$cNO_3^-$, dust-$cNO_3^-$, and $cNO_3^-$ accompanied by $cNH_4^+$ ($cNH_4^+$ term), respectively. The MRA results explained the observed seasonal changes in dust-$cNO_3^-$ and indicated that the dust-acid uptake scheme reproduced the observed dust-nitrate levels even in winter. The annual average contributions of each component were 43% (sea salt-$cNO_3^-$), 19% (dust $cNO_3^-$), and 38% ($cNH_4^+$ term). The MRA dust-$cNO_3^-$ component had a



high value during the dust season, and the sea salt component made a large contribution throughout the year. During the winter, $cNH_4^+$ term made a large contribution. The model did not include aerosol microphysical processes (such as condensation and coagulation between the fine anthropogenic aerosols $NO_3^-$ and $SO_4^{2-}$ and coarse particles), and our results suggest that inclusion of aerosol microphysical processes is critical when studying observed $cNO_3^-$ formation, especially in
winter.

## 1 Introduction

Long-range trans-boundary transport of dust and pollutants in East Asia, and their complex interactions, are an important environmental issue due to the recent rapid economic developments and changes in these areas (e.g., Lawrence and Lelieveld,
2010, Li *et al.* 2017, Zhang *et al.*, 2017). Because of the heavy air pollution occurring in China, a great deal of research focuses mainly on fine-mode aerosols (i.e., $PM_{2.5}$, particle aerodynamic diameter < 2.5 μm). NOx emissions have been increasing rapidly over the past decade (e.g., Richter *et al.*, 2002, Irie *et al.* 2016), and long-range nitrate transport is becoming increasingly important for regional nitrogen budget studies (e.g., Oita *et al.*, 2016, Itahashi *et al.*, 2016, 2017; Uno *et al.*, 2017a,b). A large proportion of nitrate exists in coarse mode ($PM_c$, particle aerodynamic diameter > 2.5 μm) due to
interactions with sea salt and mineral dust. The formation of nitrate on dust aerosols has been clearly observed using scanning electron microscopy (SEM), both in laboratory experiments and using field measurements (Li and Shao, 2009). Acid uptake of pollutants over sea salt and dust surfaces is important when evaluating coarse $NO_3^-$, as it modifies the chemical lifetime of nitric acid (including atmospheric loading and deposition). Continuous measurement of both fine- and coarse-mode aerosol compositions (including $NO_3^-$) is critical for achieving a complete understanding of the fate of air
pollutants and changes in the Asian atmospheric environment (Li *et al*, 2012, Pan *et al.*, 2017, Wang *et al.*, 2017a).
Ground-based and airborne aerosol observations have been studied to determine the physics and chemistry of high-concentration events (e.g., Huebert *et al.*, 2003; Jacob *et al.,* 2003, KORUS-AQ, 2016), but the duration of these observational campaigns is typically less than 1 month, which is insufficient for studying seasonal variation. Monitoring projects conducted by the Acid Deposition Monitoring Network in East Asia (EANET; 2014) and Asian Dust and Aerosol
Lidar Observation Network (AD-Net Lidar; Sugimoto *et al.*, 2008), among others, have been accumulating observational data for over 10 years, but detailed data on hourly aerosol compositions are rarely captured. Because seasonal changes in Asian monsoons play an important role in the patterns and frequency of long-range pollutant transport; therefore, long-term aerosol observations (over at least 1 year), including of aerosol composition and with a high time resolution, are required, but no such detailed observational studies have been undertaken to date.
We have made long-term synergetic observations of the behaviors of aerosols around the Chikushi Campus of Kyushu University, located in the suburbs of Fukuoka City (33.52°N, 130.47°E), since October 2013 (Pan *et al.*, 2015, 2106; Itahahsi *et al.*, 2017; Uno *et al.*, 2017a,b; Osada *et al.*, 2016). We used a state-of-the-art aerosol observation instrument to measure



both fine- and coarse-mode aerosols. In this study, we report seasonal variation in both fine- and coarse-mode atmospheric aerosols based on long-term synergetic aerosol observations made at 1 h intervals in Fukuoka, Japan, from August 2014 to October 2015. During this period, several yellow sand and heavy pollutant transport episodes were observed. This paper reports on the major characteristics of anthropogenic aerosols and long-range dust transport, based on observations and

chemical transport model (CTM) analysis. We also examined seasonal variations in fine- and coarse-mode long-range nitrate transport.

This report is structured as follows: Section 2 documents the observational dataset, Section 3 describes the CTM simulation in detail, Section 4 discusses temporal variations at observation sites and the model results, and Section 5 provides a summary and conclusions.

**2 Observation**

**2.1 Aerosol Chemical Speciation Analyzer and NHx**

A continuous dichotomous Aerosol Chemical Speciation Analyzer (ACSA-12 Monitor; Kimoto Electric Co., Ltd., Osaka, Japan), was utilized to measure $PM_{10}$ (particulate matter < 10 μm in diameter) and $PM_{2.5}$ (particulate matter < 2.5 μm in diameter) with high temporal resolution (Kimoto *et al*., 2013). Particulate matter (PM) was collected on a tape filter made of

Teflon (PTFE). Hourly observations were conducted to monitor $SO_4^{2-}$, $NO_3^-$, optical black carbon, and water-soluble organic compounds (WSOC) at Fukuoka. The mass concentrations of PM were determined using the beta-ray absorption method. The ACSA-12 measured $NO_3^-$ and WSOC by an ultraviolet absorption-photometric method, and $SO_4^{2-}$ by turbidimetry, after addition of $BaCl_2$ to form $BaSO_4$ and polyvinyl pyrrolidone as a stabilizer. The analytical period was within 2 h of sampling; therefore, the volatilization of particulate $NH_4NO_3$ after collection was regarded as minimal compared with the traditional

filter-pack observation method. ACSA has been tested previously (Osada et al., 2016) and used to identify aerosol chemical compositions at Fukuoka (Pan *et al*., 2016, Uno *et al*., 2017a, 2017b). It should be noted that ACSA-12 measures both fine- and coarse-mode aerosols simultaneously and is an important for mass budget studies and evaluation of CTMs.

The behaviors of $NH_3$ and $NH_4^+$ are also important because they are the counter-ions for $SO_4^{2-}$ and $NO_3^-$. Concentrations of gaseous $NH_3$ and $NH_4^+$ in fine particles were measured with a semi-continuous microflow analytical system (Kimoto Electric

Co. Ltd., MF-NH3A, Osada *et al.*, 2011). Two inlet lines were used to differentiate the total amounts of NHx and particulate $NH_4^+$ after gaseous $NH_3$ was removed using a phosphoric acid-coated denuder from the sample air stream. The cut-off diameter of the inlet impactor was about 2 μm (which is smaller than the ACSA $PM_{2.5}$ cut-off). Secondary inorganic aerosols ($SO_4^{2-}$, $NO_3^-$, and $NH_4^+$; SNA) were fully observed using our synergetic monitoring system.





## 2.2 Denuder-filter pack method

During our observation period from August 2014 to October 2015, we conducted denuder-filter (D-F) pack measurements at Fukuoka. An annular denuder–multi-stage filter sampling system was used for $HNO_3$ and size-segregated aerosol sampling. The sampling interval was 6−8 h for intensive observation and 1−2 days for regular observation. At the inlet, coarse-mode aerosols were removed by Nucleopore membrane filters (111114; Nomura Micro Science Co., Ltd., Atsugi, Japan [pore size = 8 μm]), and then gas-phase $HNO_3$ was collected with the annular denuder (2000-30x242-3CSS; URG Co., Chapel Hill, NC, USA) coated with NaCl (Perrino *et al.*, 1990). Fine-mode aerosols were collected with a PTFE filter (J100A047A; ADVANTEC, Tokyo, Japan [pore size = 1 μm]), and a nylon filter (66509; Pall Co., Port Washington, NY, USA) captured volatized nitrates from the PTFE filter (Appel *et al.*, 1981; Vecchi *et al.*, 2009). The sample air flow rate was 16.7 L min$^{-1}$ (1 atm, 25°C). Under these conditions, the aerodynamic diameter of the 50% cut-off for the Nucleopore filter was approximately 1.9 μm (John *et al.*, 1983). The samples were analyzed by ion chromatography (IC). Fine-mode aerosols (< 1.9 μm in diameter) were underestimated by the D-F pack compared with ACSA measurement, and coarse-mode aerosols (> 1.9 μm, with no upper limit) were overestimated by the D-F pack compared with the ACSA $PM_c$ measurements (2.5−10 μm) due to the a difference in cut-off diameter between the methods (Osada et al., 2016). Details of the ACSA data comparison and validation were reported previously by Osada *et al.* (2016).

## 3 Numerical modeling

We used the 3-D Goddard Earth Observing System chemical transport model (GEOS-Chem) (ver. 09-02) (Bey *et al.*, 2001, Park *et al.*, 2004, Fairlie *et al.*, 2007, 2010). The model was run with the full GEOS-Chem NOx-Ox-VOC-HOx-CO chemistry option to simulate the formation of aerosols, including mineral dust, sea salt, and secondary inorganic aerosols. We also modeled the emission/transport of primary black carbon (BC) and organic carbon (OC). However, detailed carbonaceous species and secondary organic aerosols (SOA) options in GEOS-Chem were not used in this study.

Dust in GEOS-Chem is classified according to four size bins (radii of 0.1−1.0, 1.0−1.8, 1.8−3.0, and 3.0−6.0 μm), based on Ginoux *et al.* (2004). The smallest size bin is further divided into 4 bins (radii 0.1−0.18, 0.18−0.3, 0.3−0.6, 0.6−1.0 μm) for optical properties and heterogeneous chemistry. This model uses the dust entrainment and deposition (DEAD) mobilization scheme from Zender *et al.* (2003), combined with the source function used in the Goddard Chemistry Aerosol Radiation and Transport (GOCART) model (Ginoux *et al.* 2001), as described by Fairlie *et al.* (2007). Several modifications to capture seasonal changes in dust source function, the dust emission fraction within dust bins, and the change in wet scavenging efficiency that was inversely determined to fully capture Asian dust (Yumimoto *et al.*, 2017) were used in this study. The general performance of our dust simulation was already reported by Yumimoto *et al.* (2017) and Uno *et al.* (2017a, b). Sea salt is distributed into two size bins (dry radii 0.01−0.5 and 0.5−8 μm). The sea salt emission scheme is based on the work of Jaeglé *et al.* (2011) and was well-validated by those authors. It includes the effects of sea-surface temperature (SST), as more sea salt is emitted in the summer.




In this study, the reactive uptake of $HNO_3$ and $SO_2$ on dust (limited by dust alkalinity), and the uptake of gas-phase $H_2SO_4$ (limited by competition with other aerosol surfaces) (Fairlie *et al.*, 2010) were used. Dust-nitrate (mainly $Ca(NO_3)_2$) was simulated based on the heterogeneous reaction between dust and nitric acid, as follows:

$$CaCO_3 + 2HNO_3 \rightarrow Ca(NO_3)_2 + H_2O + CO_2 \quad \text{(R1)}$$

As described by Fairlie *et al.* (2010), the first-order reactive uptake rate constant (k) is calculated according to the uptake coefficient ($\gamma$) and surface area density of dust particles (A) using the following equation, suggested by Jacob (2000):

$$-\frac{dC}{dt} = \left[ \frac{r}{D_g} + \frac{4}{c\,\gamma} \right]^{-1} A\,C = k\,C \quad (1)$$

where $C$ is the concentration of gas uptake (e.g., $HNO_3$), $r$ is the aerosol particle radius, $D_g$ is the molecular diffusion
coefficient, and $c$ is the mean molecular speed. The uptake coefficient depends on the ambient relative humidity (RH), and we used the RH-dependent function in Fig. 1 of Fairlie *et al.* (2010). More details on dust-nitrate formation from a heterogeneous reaction can be found in Fairlie *et al.* (2010). A similar heterogeneous reaction to that described by Eq. (1) for sea salt was also included in our calculation.

The model used the assimilated meteorological fields from the GEOS of the NASA Global Modeling and Assimilation
Office (GMAO). The model has a horizontal resolution of 2° × 2.5° for global runs, and 0.5° × 0.667° for Asian one-way nesting runs (11°S−55°N, 70−150°E,), both having 47 vertical levels from the surface to 0.01 hPa. The lowest model layer thickness was approximately 130 m. We used anthropogenic emissions data from the Emission Database for Global Atmospheric Research (EDGAR) (Olivier and Berdowski, 2001) for the global domain and the Regional Emission Inventory in Asia (REAS) (ver. 2.1) for the Asian domain, as reported by Kurokawa *et al.* (2013). The REAS $NH_3$ emissions were
modified to include seasonal variations in Asia based on Huang *et al.* (2012), and further changes in winter emissions recommended by Xu *et al* (2015). The model simulation was conducted from the beginning of December 2013 to the end of October 2015, and results from the first 8 months were used to train the model. Other basic numerical settings were as reported in Uno *et al.* (2017a, b).

To investigate whether domestic or transboundary air pollution is dominant in Japan, we also performed a sensitivity
simulation. Because the quantity of emissions from China was larger than that from Japan, to avoid large nonlinearities in the atmospheric concentration response to emissions variation (e.g., Itahashi *et al.*, 2015), a sensitivity simulation was designed to include a 20% reduction in anthropogenic emissions from Japan (defined as JOFF20%). Based on differences between the baseline simulation (CNTL) and the JOFF20% simulation, the domestic contribution from Japan (JOFF) was estimated using a multiple of 5 for differences in CNTL and JOFF20% experiments. We also made a volcanic $SO_2$
sensitivity simulation without volcanic $SO_2$ emissions (VOFF) because the $SO_2$ to $SO_4^{2-}$ formation can be assumed to be linear.





Based on the analysis of PM concentrations, $PM_{2.5}$ and $PM_{10}$ were calculated by summing the individual aerosol ($SO_4^{2-}$, $NO_3^-$, $NH_4^+$, BC and OC), dust, and sea salt components of the model. Hereafter, we denote modeled "ammonium" nitrate (i.e., $NH_4NO_3$) as $A-NO_3^-$, dust-nitrate as $D-NO_3^-$, sea salt nitrate as $SS-NO_3^-$, and their sum as simply $NO_3^-$ (= A- + D- + $SS-NO_3^-$).

## 4 Results and Discussion

### 4.1 Meteorological variation and sea salt

Fig. 1 shows the location of the observation sites, Fukuoka and Mt. Sakurajima volcano, and the anthropogenic $SO_2$ emission distribution used in the model calculation. Fig. 2 shows (a) the daily mean and maximum wind speed, (b) the daily mean temperature, RH, and precipitation in Fukuoka, as observed by the Japan Meteorological Agency, and (c) the observed coarse-mode $Na^+$ by D-F, and GEOS-Chem-simulated coarse-mode sea salt (converted to $Na^+$ concentration).

Japanese weather is controlled by changes in Asian monsoons. Under summer monsoon conditions (covered by S-SE wind from the hot and moist air mass of the Pacific High), air temperature and RH are at their maximum, while during the winter monsoon (N-NW continental cold air outflow), low temperatures and less precipitation are observed in Fukuoka. Maximum wind speeds in excess of 10 m s$^{-1}$ in summer−fall show the effects of typhoons (Fig. 2a). The precipitation difference between August 2014 and August 2015 is important for examining differences in the $NH_3$ concentration (August 2014 had more rainfall than August 2015).

The time variation of modeled coarse-mode $Na^+$ generally agreed well with observations, except during typhoon events. There were five typhoons between July and September 2015, and the modeled $cNa^+$ was very high compared with observations. This indicated that the sea salt emission scheme for very strong wind conditions was overestimated. We also observed high precipitation during the typhoon events. Another important difference is that the typhoon in 2014 occurred in the fall (September−October), after the SST had cooled, while in 2015 it occurred in the summer (July−August) when the SST was warm and more sea salt was emitted.

### 4.2 Time-series variations in $PM_{2.5}$, $PM_{10}$, black carbon, and CO

Fig. 3 (a)–(d) show the time variations (1h) in $PM_{2.5}$, $PM_{10}$ optical BC (ACSA-12) and CO observations (Thermo Model 48) at Fukuoka. The figures also include the corresponding model values (2 h).

The observed $PM_{2.5}$ clearly exhibited frequent intermittent peaks from winter to early spring, but the frequency decreased after April according to the frequency of "polluted" cold air outbreaks from the Asian continent. The modeled $PM_{2.5}$ reproduced most of the observed variation. The modeled $PM_{2.5}$ value corresponded to approximately 58% of the observed $PM_{2.5}$ ($PM_{2.5\_model}$ = 0.58 ($PM_{2.5\_obs}$) + 1.16 µg m$^{-3}$, $R$ = 0.72) because the observation data included aerosol compositions that were not incorporated into the model (e.g., secondary organic aerosols, many trace metals, and others).



The observed $PM_{10}$ included coarse aerosols (e.g., dust and sea salt), and peaks corresponded to major dust events. Symbols A–G are the major dust events that occurred during our observation period (Uno *et al.*, 2017b). The time variation of modeled $PM_{10}$ was in good agreement with observations, but was consistently underestimated for the same reasons as for $PM_{2.5}$ (i.e., more coarse aerosols were included in the observed data than in the model) and showed some uncertainty for

large dust concentrations (Uno *et al.*, 2017b).

Modeled BC corresponded to approximately 33% of the observed BC ($BC_{model}$ = 0.33 ($BC_{obs}$) + 0.23 μg m$^{-3}$, $R$ = 0.35). Modeled BC was systematically underestimated from July to December, but was at reasonable levels during the winter. One possible explanation is underestimation of Japanese BC emission intensity (e.g., Itahashi *et al.*, 2017).

Modeled CO concentrations reproduced seasonal variation very well (winter = high and summer = low), but the modeled

concentration level was 43% of the observed level ($CO_{model}$ = 0.43 ($CO_{obs}$) + 21.2 ppb, $R$ = 0.69). This may be due to underestimation of globally assumed background levels of CO. Observed CO and BC were highly correlated ($R$ = 0.61), and a high CO concentration can be considered a product of long-range transport (LRT). From these comparisons of PM, BC, and CO, we found that the model simulation captured the major LRT outflow well, and can be used for detailed analysis of aerosol transport.

### 4.3 Daily and monthly variation in PM and aerosol compositions

Fig. 4 shows the daily average aerosol composition for (a) fine $SO_4^{2-}$, (b) fine $NO_3^-$, and (c) coarse $NO_3^-$. ACSA observations are indicated in blue; D-F pack observations are indicated by the red line (spike-like high peaks during the dust event are due to large particles [>10 μm, which is the cut-off in ACSA sampling]), and the gray shading indicates the model control experiment (CNTL). Yellow shading in Fig. 4(c) indicates the modeled dust-coarse $NO_3^-$ (D-c$NO_3^-$) fraction. The modeled

c$NO_3^-$ is the sum of dust-nitrates and sea salt-nitrates.

Fig. 5 shows the monthly average (a) $PM_{2.5}$, (b) fine- and coarse-mode $SO_4^{2-}$, (c) fine and coarse $NH_4^+$, and (d) $NH_3$ gas. Box-whisker plots show the observed fine aerosol levels (10$^{th}$, 25$^{th}$, 50$^{th}$, 75$^{th}$ and 90$^{th}$ percentile values are marked). The observed average monthly coarse-mode aerosols are shown by the red dashed line. The modeled average monthly fine aerosol concentration for the CNTL is depicted by the straight black line, and by the black dashed line for the JOFF.

Fig. 6 shows the (a) $HNO_3$, (b) total $NO_3^-$, (c) coarse $NO_3^-$, and (d) fine $NO_3^-$ levels. Observed values are shown in box-whisker plots. Modeled monthly averages are also shown. The JOFF model sensitivity experiment is denoted by the black dashed line in Fig. 6a, 6b, and 6d.

### 4.3.1 Fine-mode $SO_4^{2-}$ and $NH_4^+$

The modeled $SO_4^{2-}$ results explained the observed time variations and corresponded to 37% of the modeled $PM_{2.5}$

concentration ($SO_4^{2-}{}_{model}$ = 0.82($SO_4^{2-}{}_{obs}$) + 0.76 μg m$^{-3}$, $R$ = 0.68; $SO_4^{2-}{}_{model}$ = 0.37($PM_{2.5\_model}$) + 0.40 μg m$^{-3}$, $R$ = 0.81). The high correlation suggests that the variations in $PM_{2.5}$ were mainly driven by f$SO_4^{2-}$, as a main component of $PM_{2.5}$ for both the observations and the model.





Fig. 4a shows intermittent high $SO_4^{2-}$ concentrations. High $SO_4^{2-}$ levels in the winter depended on the frequency of LRT from the Asian continent, but high $SO_4^{2-}$ levels are usually observed in summer (see also **Fig. 5b**). This is related to the meteorological conditions, i.e., high RH in summer. Fig. 4a shows high $SO_4^{2-}$ levels from the end of July to early August 2015 (designated SF1 in the figure). These high $SO_4^{2-}$ concentrations were due to Japanese domestic emissions and volcanic

$SO_2$ emissions (see the appendix for a more detailed analysis).

The model underestimated $SO_4^{2-}$ levels from December to January. Similar underestimation was observed in other CTM applications (Wang *et al.,* 2017b; Itahashi *et al.*, 2017), and in the observations of He *et al.* (2014).

Based on Fig. 5, $SO_4^{2-}$ and $NH_4^+$ levels are reasonably well predicted by the model. The JOFF sensitivity analysis showed that the contribution from Japanese emissions was small and the majority of $SO_4^{2-}$ came from LRT from outside of Japan

(annual mean Japanese domestic contribution = 15%). For $NH_4^+$, we found that the LRT contribution was large until April, after which the Japanese contribution increased as $NH_3$ emissions increased (annual mean Japanese domestic contribution = 27%).

Coarse mode $SO_4^{2-}$ was about 0.1−0.2 of fine $SO_4^{2-}$ (monthly average relationship: $cSO_4^{2-} = 0.185 fSO_4^{2-} – 0.052 \mu g\ m^{-3}$, $R =$ 0.66). The increases in coarse $SO_4^{2-}$ contribution cannot be ignored in winter. Modeled $cSO_4^{2-}$ levels were very small

compared to observed levels and are not shown in the figure.

### 4.3.2 Fine-mode $NO_3^-$

The modeled fine $NO_3^-$ reproduced the major time variations seen in the observed levels. $NO_3^-$ levels are higher during the winter and lower during the summer, exhibiting clear seasonal changes. The winter (DJF) $fNO_3^-$ average was 2.62 ± 1.64 µg

$m^{-3}$. The daily maximum reached up to 6−9 µg $m^{-3}$ and was sometimes higher than the $SO_4^{2-}$ level, with the same timing of peaks as for $SO_4^{2-}$. This indicated that $fNO_3^-$ were also controlled by LRT from the Asian continent (as discussed by Itahashi *et al*., 2017; Uno *et al*., 2017b). The JOFF sensitivity analysis also confirmed that the majority of $fNO_3^-$ can be considered due to LRT from outside Japan during the winter. From mid-May−October, the Japanese contribution becomes dominant (annual mean Japanese domestic contribution = 53%).

In the summer, $fNO_3^-$ is minimal because the $NH_4NO_3$ equilibrium between $NH_3$ and $HNO_3$ shifts to the gaseous phase under warm conditions. This phase shift is a function of temperature and RH, and it moves to an aerosol phase at cold temperatures. The monthly average $fNO_3^-$ from June to September was 0.68 ± 0.34 µg $m^{-3}$. It should be noted that $NO_3^-$ measurement by ACSA was finished within 2 h of PTFE sampling, so any artifact due to volatilization of $NO_3^-$ from the Teflon filter surface was small (e.g., Osada *et al.*, 2016). As described in section 2, D-F measurement used the nylon backup file to catch the

volatilized $HNO_3$ from the Teflon filter surface. Thus, both ACSA $fNO_3^-$ and D-F $fNO_3^-$ measurements showed consistent concentrations during the summer (average = 0.43−0.68 µg $m^{-3}$; see Fig. 4b.)





### 4.3.3 Coarse-mode $NO_3^-$

Fig. 4c shows that the coarse-mode $NO_3^-$ had a distinct peak value during dust events A−G (D-F $cNO_3^-$ levels are higher than ACSA $cNO_3^-$ levels due to different upper cut-off limits). Modeled $cNO_3^-$ levels were higher (see Fig. 4c and 6c) during the fall (October to November) and during the dust seasons (typically from February to June).

Fig. 6c shows the clear seasonal cycle of modeled $cNO_3^-$. The modeled results showed that dust-nitrate concentrations increased during dust episodes. Modeled sea salt-nitrate was consistently in the order of 0.5−1.0 µg m$^{-3}$ as the baseline $cNO_3^-$, and it was the dominant except during the dust season (showing good agreement with observed values).

Table 1 summarizes the comparison of $cNO_3^-$ levels. Except for January to June, sea salt-$NO_3^-$ was the dominant within $cNO_3^-$, and the model results were in good agreement with actual observations. From January to June, the ratio of D-$cNO_3^-$ to

10 SS-$cNO_3^-$ ≈ 1:1 and the modeled total $cNO_3^-$ corresponded to two-thirds of the observed total $cNO_3$- (annual average ratio of D-$cNO_3^-$ to SS-$cNO_3^-$ ≈ 1 : 2.4). From January−June 2015, we observed several dust events (designated A−G). Uno *et al.* (2017b) described the typical onset of dust events B and C, and pointed out underestimation of the modeled $cNO_3^-$ during these cold dust cases. In this paper, we examine dust event G as a case study of warm weather dust acid uptake validation.

Fig. 7 shows the daily changes in dust (colored region) and total D-$cNO_3^-$ (contoured area) horizontal distributions from (a)

15 June 12 to (b) June 13, and the Hybrid Single Particle Lagrangian Integrated Trajectory (HYSPLIT) model back trajectory starting from Fukuoka. It also shows a comparison of modeled and observed (c) fine- and (d) coarse-mode $NO_3^-$. The dust transport path was very similar to those of dust events B and C, as described by Uno *et al.* (2017b). For dust events B and C (see Fig. 4c), modeled $cNO_3^-$ was underestimated, but for dust event G, modeled $cNO_3^-$ levels were in good agreement with observed levels. Modeled results show that D-$cNO_3^-$ mainly formed over the Yellow Sea and the East China Sea. This

indicates that the underestimation of dust-$NO_3^-$ over the winter was independent from the dust acid uptake scheme. These results indicate the importance of further studies on the mechanism of $cNO_3^-$ formation.

### 4.3.4 Monthly variation in total $NO_3$ and $NH_x$

Fig. 8 shows the observed monthly variation in (a) $NH_x$, (b) $fNO_3^-$, $cNO_3^-$, and $HNO_3$, and (c) the relative mass fraction of $NO_3^-$ and $HNO_3$. From this figure, $fNO_3^-$ clearly showed the highest concentration in winter, while $HNO_3$ increased in

summer ($HNO_3$ accounted for 30−40% of total $NO_3^-$); this was due to the change in thermal equilibrium between gases and particles. Notably, $cNO_3^-$ was always higher than $fNO_3^-$ ($cNO_3^-$ made up 27−55% of the total $NO_3^-$, and it exceeded 45% in winter). We observed 0.5−1.0 µg m$^{-3}$ (0.2−0.4 ppb) of $HNO_3$ even in winter.

The observations showed that fine $NH_4^+$ is higher throughout the year, and $NH_3$ gas is also higher even in winter. This high $NH_3$ concentration may be influenced by local agriculture and poultry farming 5−10 km south of the observation site. The

30 high $NH_3$ concentration (see Fig. 5d) in August 2015 (four times higher than August 2014) was due to differences in temperature and precipitation. The annual average $cNH_4^+$/total $NH_x$ ratio was 10%, but it increased to 15% (JFM average). This indicates that the $cNH_4^+$ counterpart in winter is important for understanding $cNO_3^-$ and $cSO_4^{2-}$.





The model results underestimated $NH_3$ and overestimated $HNO_3$; however, the modeled $fNO_3^-$ and $fNH_4^+$ levels were in good agreement with the observed values (compared with Fig.4a,c,d). The $NH_3$ emissions inventory is at 25 km resolution, and does not include the impact of local agriculture and poultry farming (which has large uncertainty); this results in the decreased bias in $NH_3$ emissions seen in our study. However, this underestimation of $NH_3$ emissions was not a critical factor in the discussion of $fNO_3^-$, because the $NH_4NO_3$ equilibrium between $NH_3$ and $HNO_3$ is given as

$$HNO_{3\,(g)} + NH_{3\,(g)} \rightleftarrows NH_4NO_{3\,(p)} \quad (R2)$$

and the equilibrium constant is proportional to the product of partial pressure $P_{NH3}$ and $P_{HNO3}$. The $HNO_3$-high (-low) and $NH_3$-low (-high) relationship may retain the same equilibrium constant, and this may be the reason for the agreement of $fNH_4^+$ and $fNO_3^-$ levels with observed values. Most of the $cNO_3^-$ formation occurred before arrival in Japan (it mainly occurred over the ocean), and was also not strongly sensitive to $NH_3$ emissions.

The present model does not include the formation scheme of $cNH_4^+$. The counterparts of $cNH_4^+$ can be $cNO_3^-$ and $cSO_4^{2-}$, and this is one of the reasons for $cNO_3^-$ underestimation by the model. It is important to describe the possible reasons for this underestimation to conduct a detailed N budget study, because the $cNO_3^-$ fraction is very large. Another interesting question is whether the modeled acid-uptake scheme can explain the observed $cNO_3^-$ levels in dust and sea salt. We used statistical analysis to investigate this.

### 4.4 Statistical analysis of coarse-mode $NO_3^-$

Major sources of $cNO_3^-$ include dust, sea salt and other sources, represented as $cNH_4^+$. We conducted multiple regression analysis (MRA) to analyze $cNO_3^-$. We included $cNa^+$, nss-$cCa^{2+}$ and $cNH_4^+$ as independent variables to describe the observed $cNO_3^-$ ($cSO_4^{2-}$ was excluded from the MRA due to its strong colinearity with $cNH_4^+$). One important point is that the observation site is surrounded by school grounds and a large city park, providing background local dust. We found that the median value of nss-$cCa^{2+}$ during the non-dust season was 0.25 µg m$^{-3}$, and the residence time of local dust coming to the observation site for producing $cNO_3^-$ was short (< 1 h). We excluded observations less than nss-$cCa^{2+}$ < 0.25 µg m$^{-3}$ from the MRA.

The result of the MRA (n = 160, nss-cCa > 0.25 µg m$^{-3}$) was

$$cNO_3^- = 0.51\ cNa^+ + 1.58\ \text{nss-}cCa^{2+} + 2.2\ cNH_4^+ + 0.19\ \text{µg m}^{-3}\ (R^2 = 0.74,\ p < 0.001)$$

where the term on the right can be considered to represent SS-$cNO_3^-$, D-$cNO_3^-$, and $cNO_3^-$ accompanied by $cNH_4^+$, respectively.

Fig. 9 shows the contribution of (a) MRA D-$cNO_3^-$ = 1.58 × nss-$cCa^{2+}$ for nss-$cCa^{2+}$ > 0.25 µg m$^{-3}$, and (b) MRA SS-$cNO_3^-$ = 0.51 × $cNa^+$. The figure also includes the GEOS-Chem modeled results (averaged over the same time period as for the D-F measurements). MRA D-$cNO_3^-$ showed good agreement with modeled D-$cNO_3^-$ levels (except for intensive dust events B and C), and this indicates that the dust acid-uptake scheme explains the observed dust-nitrate formation (underestimation for B and C comes from the underestimation of modeled dust concentrations). As previously pointed out, modeled SS-$cNO_3^-$



was overpredicted during the summer typhoon period; aside from these periods, good agreement was found with sea salt $cNO_3$ (Fig. 9b).

Fig. 10 shows the monthly averaged MRA results and clearly explains the observed seasonal variations. **Table 1** summarizes each term and shows the comparison with CTM. The average annual contribution of each term was 40% for SS-$cNO_3^-$, 20%
for D-$cNO_3^-$ and 40% for $cNH_4^+$ term. The D-$cNO_3^-$ value was high during the dust season, and the sea salt component made the largest contribution throughout the year. The $cNH_4^+$ term made a large contribution during the winter, and one possible reason for this is the condensation/coagulation of small $NH_4NO_3$ and $(NH_4)_2SO_4$ particles onto large particles (e.g., sea salt and dust). The annual average D-$cNO_3^-$ to SS-$cNO_3^-$ ratio in MRA was 1 : 2, which is close to the modeled ratio (1 : 2.4), as shown in Table 1.

Our results suggest that inclusion of aerosol microphysical processes (such as condensation and coagulation of the fine anthropogenic aerosols $NO_3^-$ and $SO_4^{2-}$ onto the coarse particles) is important for exploring the observed $cNO_3^-$ concentrations. A CTM incorporating advanced particle microphysics (APM) is one potential option (Yu and Luo, 2009). Such a modeling approach, incorporating interactions with mineral dust and sea salt, has not yet been fully explored in East Asia and is a future research direction.

**5 Conclusions**

Long-term synergetic fine- and coarse-mode aerosol observations were analyzed at 1 h intervals in Fukuoka, Japan, from August 2014 to October 2015. A GEOS-Chem chemical transport model including dust and sea salt acid uptake processes was used for detailed analysis of observation data to understand the effects of LRT from the Asian continent. The findings from this study can be summarized as follows:

1) Continuous measurement of $SO_4^{2-}$, $NO_3^-$, and optical BC with the ACSA-12 monitor, and of $NH_4^+$ and $NH_3$ with a semi-continuous microflow analytical system (MF-NH3A) in addition to D-F pack/IC analysis, established long-term synergetic fine- and coarse-mode aerosol data base in Fukuoka, Japan.

2) During the observation period, several Asian dust and long-range anthropogenic aerosol transport events were observed,
and a numerical model generally explained the observed time variations. A dust concentration by dust emission inversion scheme effectively reproduced the major dust onset.

3) Numerical results explained the seasonal changes in fine aerosols ($SO_4^{2-}$, $NO_3^-$, and $NH_4^+$), and the impact of LRT, and a sensitivity analysis excluding Japanese domestic emissions revealed the LRT fraction (85% for $fSO_4^{2-}$, 47% for $fNO_3^-$ and 73% for $fNH_4^+$).

4) Observational data for $HNO_3$, $fNO_3^-$, and $cNO_3^-$ confirmed that there was a large amount of coarse $NO_3^-$ comprising 40−55% of total nitrate during the winter; meanwhile, $HNO_3$ gas made up 40% of total nitrate in summer, and fine $NO_3^-$




peaked during the winter. A numerical model reproduced the seasonal variation in fine $NO_3^-$. For coarse $NO_3^-$, modeled $cNO_3^-$ was in good agreement with observations between July and November (mainly coming from sea salt-$NO_3^-$).

5) Large-scale dust-nitrate outflow from China to Fukuoka was confirmed in all dust events. During the winter, modeled coarse $NO_3^-$ was underestimated compared to the observed levels.

6) The coarse $NO_3^-$ underestimation was examined statistically using multiple regression analysis (MRA). We included $cNa^+$, $nss-cCa^{2+}$ and $cNH_4^+$ as independent variables to describe the observed $cNO_3^-$; these terms can be considered to represent $SS-cNO_3^-$, $D-cNO_3^-$, and $cNO_3^-$ accompanied by $cNH_4^+$, respectively. The MRA results showed a high correlation, $R^2 = 0.74$, and clearly explained the observed seasonal changes. The annual average contributions of the component were 43% ($SS-cNO_3^-$), 19% ($D-cNO_3^-$), and 38% ($cNH_4^+$ term). The $D-cNO_3^-$ component had the largest

value during the dust season, and the sea salt component made a large contribution throughout the year. $cNH_4^+$ made a large contribution during the winter; one possible reason for this was the condensation/coagulation of small $NH_4NO_3$ and $(NH_4)_2SO_4$ particles onto large particles (e.g., dust and sea salt).

The present model did not include aerosol microphysical processes (such as condensation and coagulation between fine

anthropogenic aerosols, such as $NO_3^-$ and $SO_4^{2-}$, and coarse particles), but our results suggest that aerosol microphysical processes are important for studying observed $cNO_3^-$ formation, especially in winter.

**Data availability.** To request observation data used in this study for scientific research purposes, please contact Itsushi Uno

at Kyushu University via email (uno@riam.kyushu-u.ac.jp). Model simulations were based on GEOS Chem, open-source and publicly available software. The model and related software can be downloaded from http://acmg.seas.harvard.edu/geos/ for registered users.

**Author contributions.** Itsushi Uno designed the synergetic observations at Chikushi Campus of Kyushu University, Japan.

Shigekazu Yamamoto and Kazuo Osada respectively carried out the ground-based ACSA and NHx-monitor observations at Fukuoka. Kazuo Osada conducted air sampling and chemical analysis for D-F pack samples during the observation period at Fukuoka. Yugo Kanaya and Xaiole Pan conducted the CO observations at Fukuoka. T. Duncan Fairlie provided GEOS Chem modeling system. Itsushi Uno, Keiya Yumimoto and Zhe Wang performed the model simulations and analysis, and prepared the paper with contributions from all coauthors.

**Competing interests.** The authors declare that they have no conflict of interest.

**Acknowledgments.** This work was supported by the Ministry of Education, Culture, Sports, Science and Technology (MEXT), the Japan Society for the Promotion of Science (JSPS) and the Grants-in-Aid for Scientific Research (KAKENHI)



program (Grant JP25220101). The authors would like to thank the developers of the GEOS reanalysis product. This work was partly funded by a collaborative research program through the Research Institute for Applied Mechanics at Kyushu University (no. 26AO-2, 27AO-6, 28AO-2). We thank Yusuke Kamiguchi at Nagoya University for D-F pack sampling and chemical analysis and Shohei Kuwahara at Kyushu University for D-F pack sampling.



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





**Appendix**

During the SF1 period (shown in Fig. 4), two typhoons passed south of Japan from east to west. Fig. A1 shows snapshots of
the trajectories and wind fields of Typhoon Halola (T12) and Typhoon Soudelor (T13), along with modeled $SO_4^{2-}$
concentrations.

$SO_4^{2-}$ in the northern part of Japan and Korea were under the control of long-range transport from China, while the southern
part of Japan was covered by east winds (due to typhoons). This indicated that the high $SO_4^{2-}$ concentrations in Fukuoka
were due to Japanese domestic emissions and the impact of Mt. Sakurajima volcano (typically August 2−6). Note that the
estimated $SO_2$ emission from Mt. Sakurajima is around $10^3$ Gg year$^{-1}$, which is almost equivalent to the level of
anthropogenic $SO_2$ emissions in Japan.

Fig. A2 shows the volcano sensitivity experiment (VOFF) for $SO_4^2$. It is clear that the impact of the volcano was higher for
the period August 2−6; during this period, $SO_4^{2-}$ contributions converted from volcanic $SO_2$ emission exceeded 50% of the
total $SO_4^{2-}$. Observations show higher values from July 27−August 1, and the model simulation failed to describe this.
Volcanic emissions are natural phenomena and have strong day-by-day variation, so this failure of the model is due to
underestimation of volcanic emissions during this period.



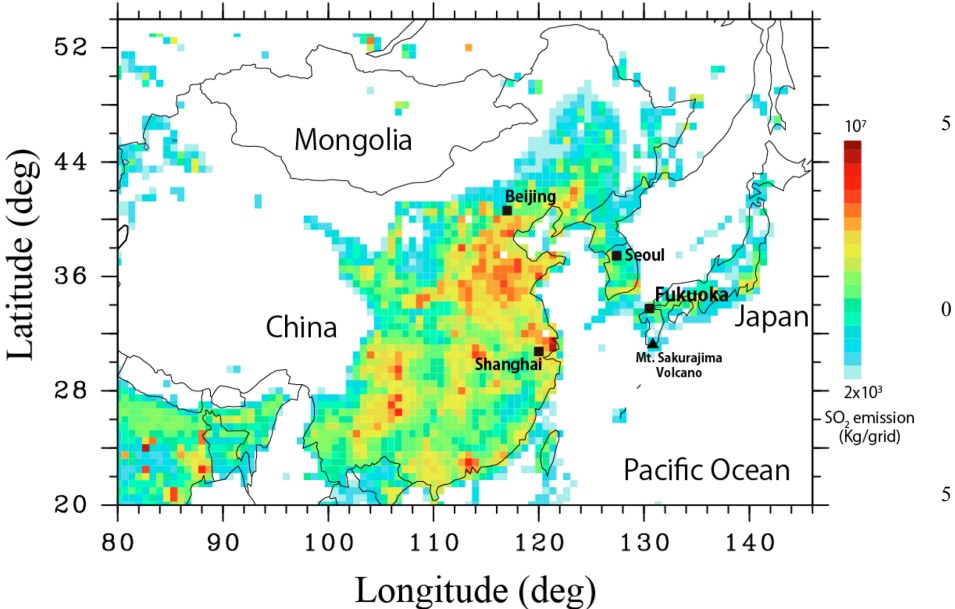

**Figure 1.** SO₂ emissions intensity (based on 2008 data) and observation sites: Fukuoka and Mt. Sakurajima volcano.




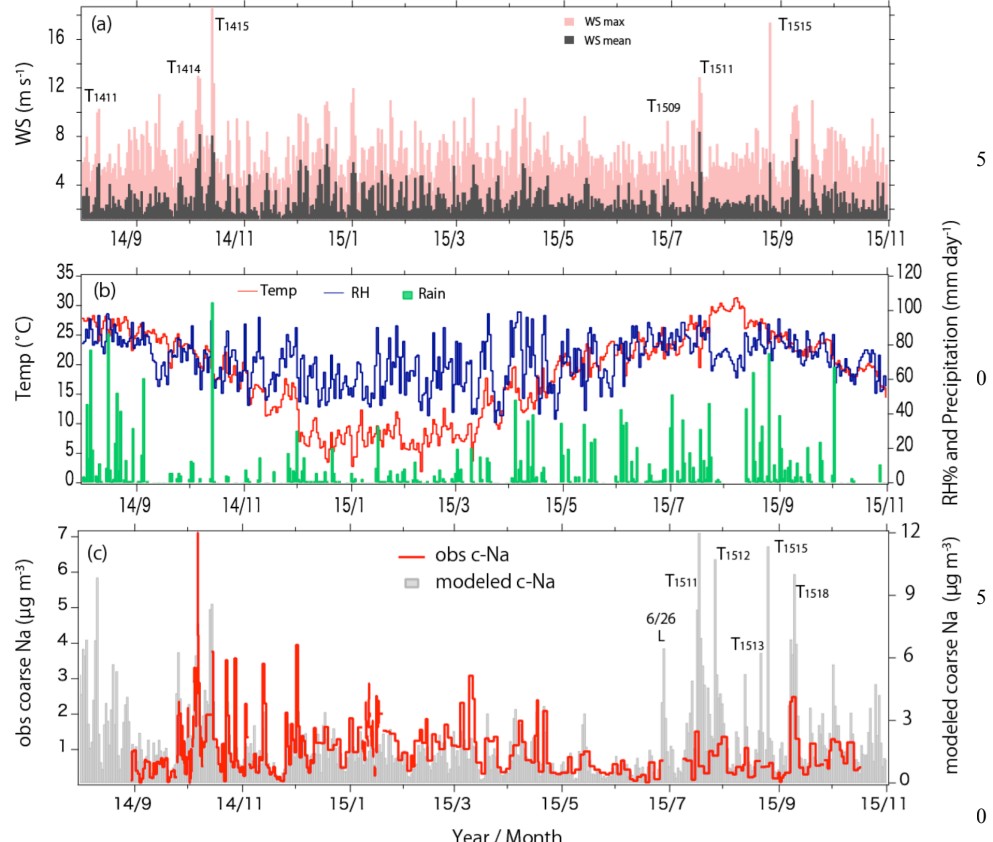

**Figure 2. (a) Wind speeds (daily mean and maximum) at Fukuoka. (b) Daily average temperature, relative humidity (RH) and precipitation. (c) Observed coarse Na$^+$ and modeled coarse Na$^+$ (converted from GEOS-Chem coarse sea-salt (SALC)). Tyyxx indicates typhoons: T1411 = Halong, T1414 = Fengshen, T1415 = Kalmaegi, T1509 = Chanhom, T1511 = Nangka, T1512 = Halola, T1513 = Soudelor, T1515 = Goni, T1518 = Etau.**



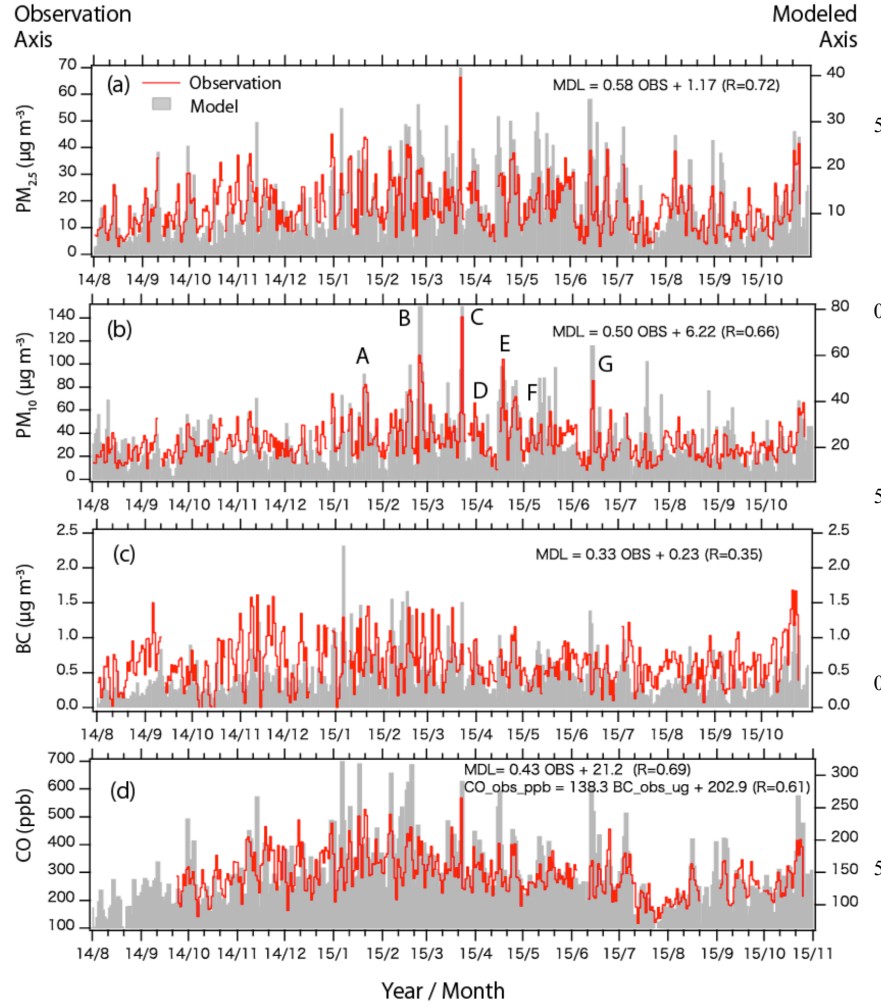

**Figure 3.** Daily average (a) particulate matter < 2.5 μm diameter (PM$_{2.5}$), (b) particulate matter < 10 μm diameter (PM$_{10}$), (c) black carbon (BC), and (d) CO. Red lines show the observed values, and gray shading indicates the numerical results. Observed data are on the left axis and numerical results are on the right axis, scaled by the regression results (except for c). Symbols A−G in (b) indicate major dust events in Fukuoka.




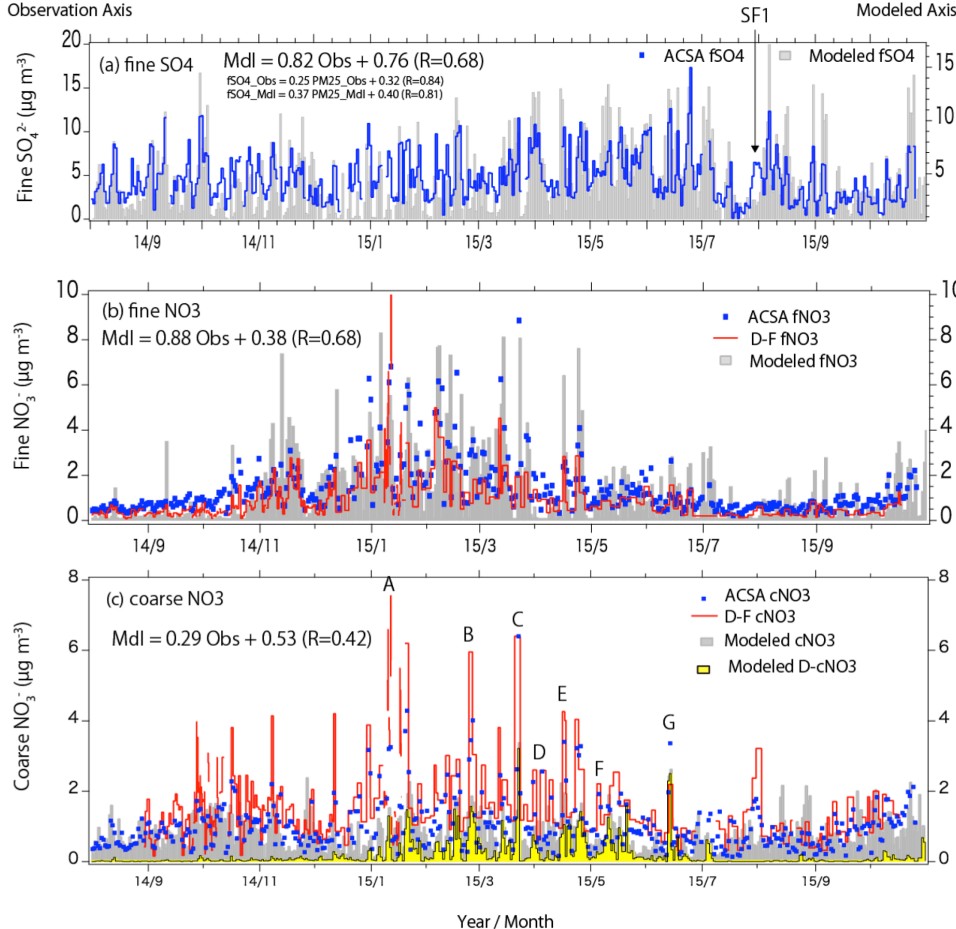

**Figure 4. Daily average variation in (a) fine SO$_4^{2-}$, (b) fine NO$_3^-$, and (c) coarse NO$_3^-$. The blue line and dots show Aerosol Chemical Speciation Analyzer (ACSA) observations, the red line indicates measurements from the denuder filter pack (D-F) method, and gray shading indicates the model simulation. D-F data are sampling period averages.**



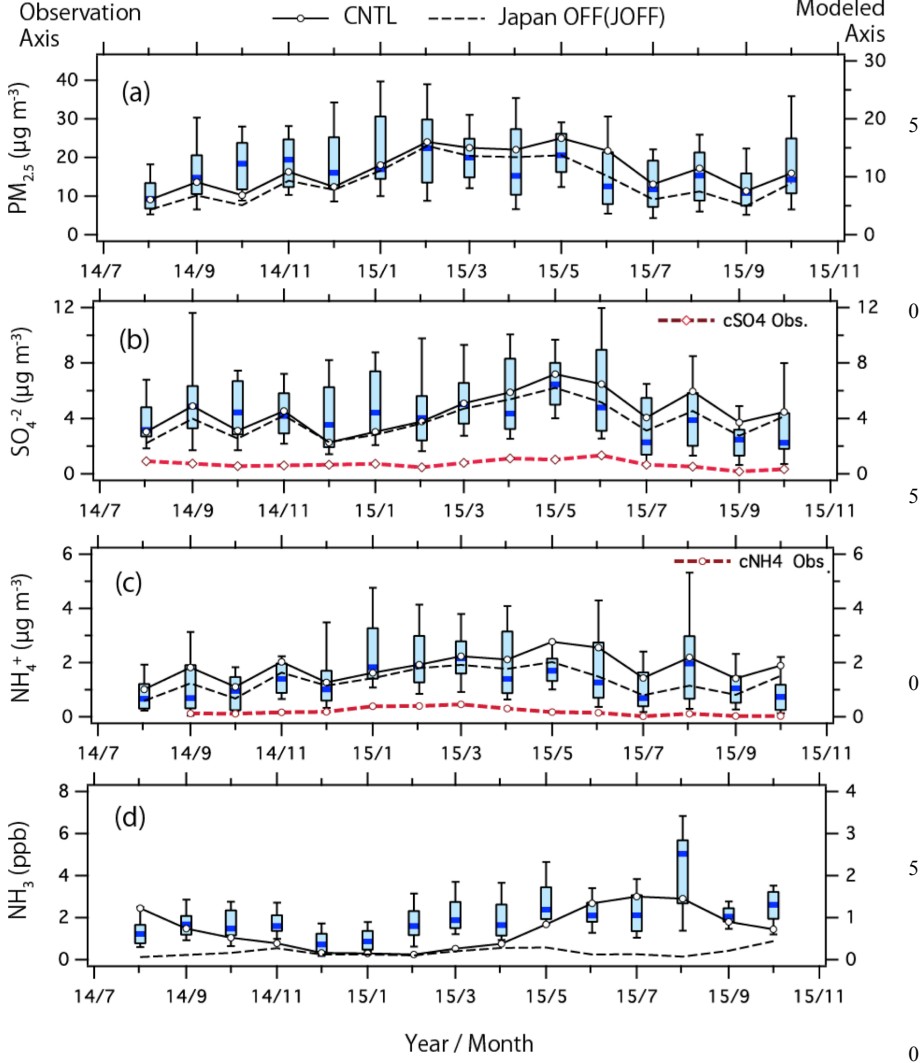

**Figure 5. Monthly average plots of (a) PM$_{2.5}$, (b) fine SO$_4^{2-}$, (c) fine NH$_4^+$, and (d) NH$_3$ gas. Box-whisker plots show observations (10$^{th}$, 25$^{th}$, 50$^{th}$, 75$^{th}$, 90$^{th}$ percentile values). Red lines in (b) and (c) show ACSA-observed coarse mode concentrations. Black lines are modeled monthly averages; the straight line indicates the baseline model (CNTL), and the dashed line indicates the domestic emissions contribution from Japan (JOFF).**





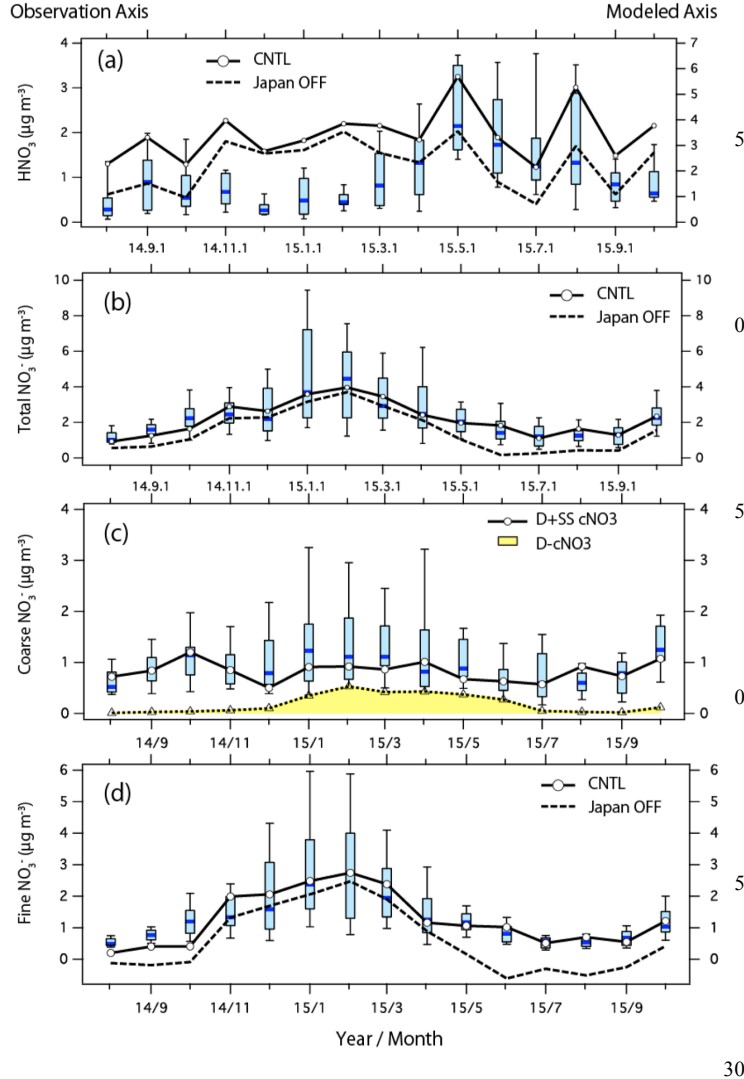

**Figure 6.** Monthly average (a) HNO₃, (b) total NO₃⁻, (c) coarse NO₃⁻, and (d) fine NO₃⁻. Box-whisker plots show observations (10th, 25th, 50th, 75th, 90th percentile values), and black lines are modeled monthly averages; the straight line indicates CNTL, and the dashed line, except in (c), indicates JOFF. For (c), the dashed line shows the modeled dust-coarse NO₃⁻ (D-cNO₃⁻). HNO₃ observations are from D-F, and NO₃⁻ observations are from ACSA.





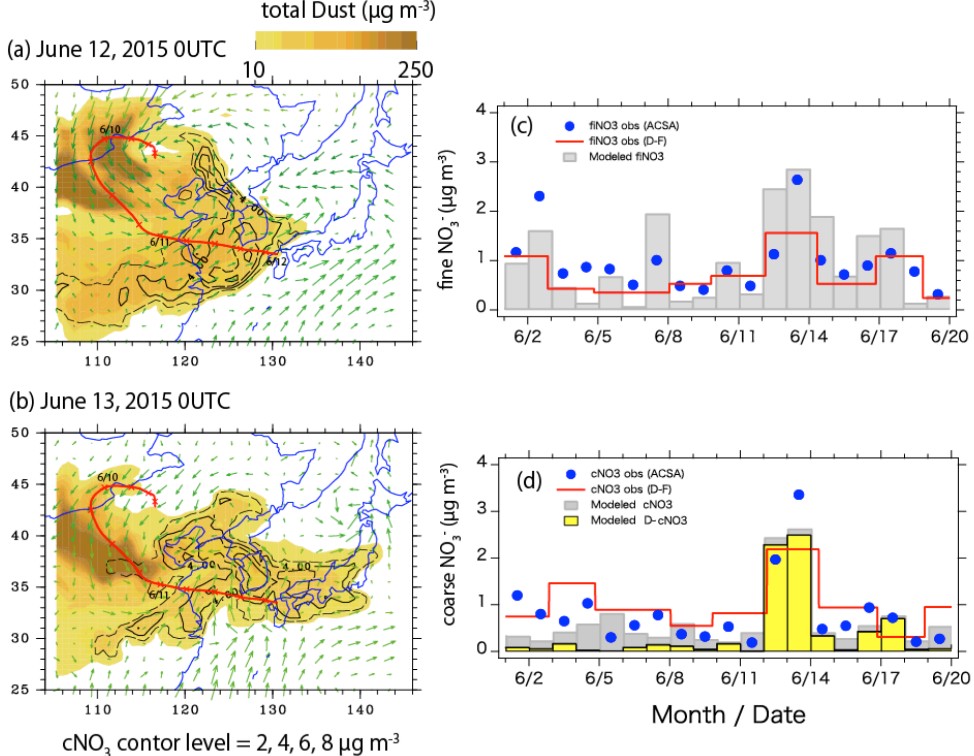

**Figure 7. Horizontal distribution of dust (sum of all dust bins) (yellow color) and fine $NO_3^-$ (contour) with wind vectors for (a) June 12, (b) June 13, and the time variation of daily average (c) fine $NO_3^-$ and (d) coarse $NO_3^-$.**




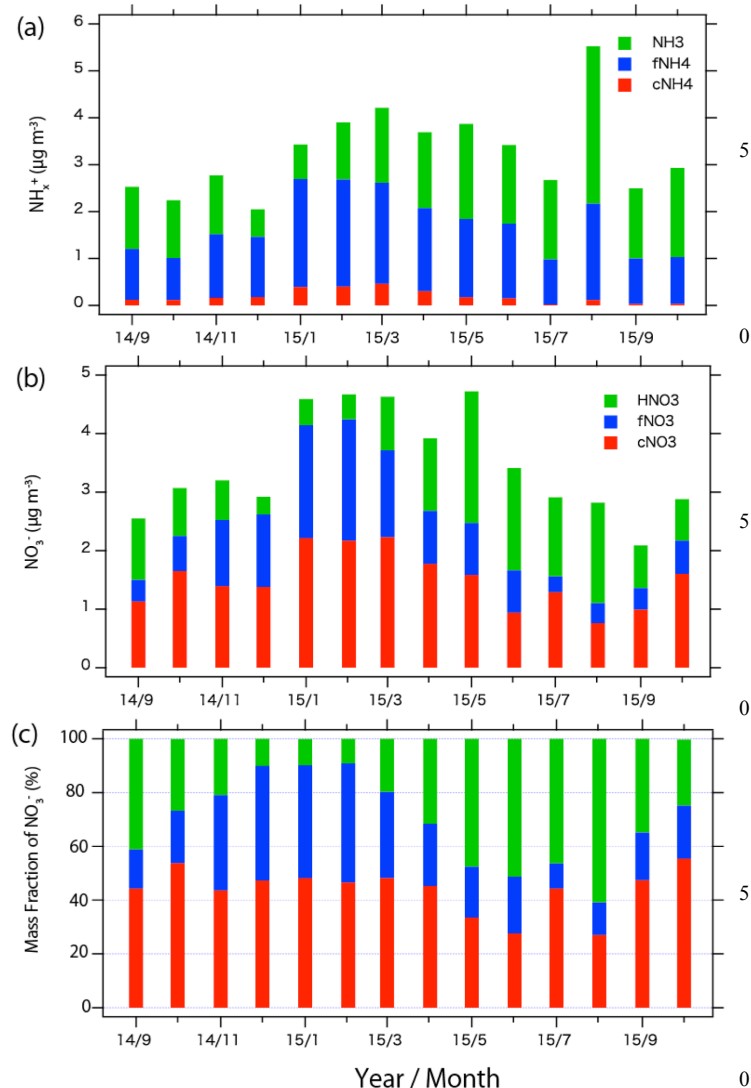

**Figure 8. Monthly average values for (a) NHx, (b) NO₃, and (c) the mass fraction (%) of NO₃, observed by D-F.**





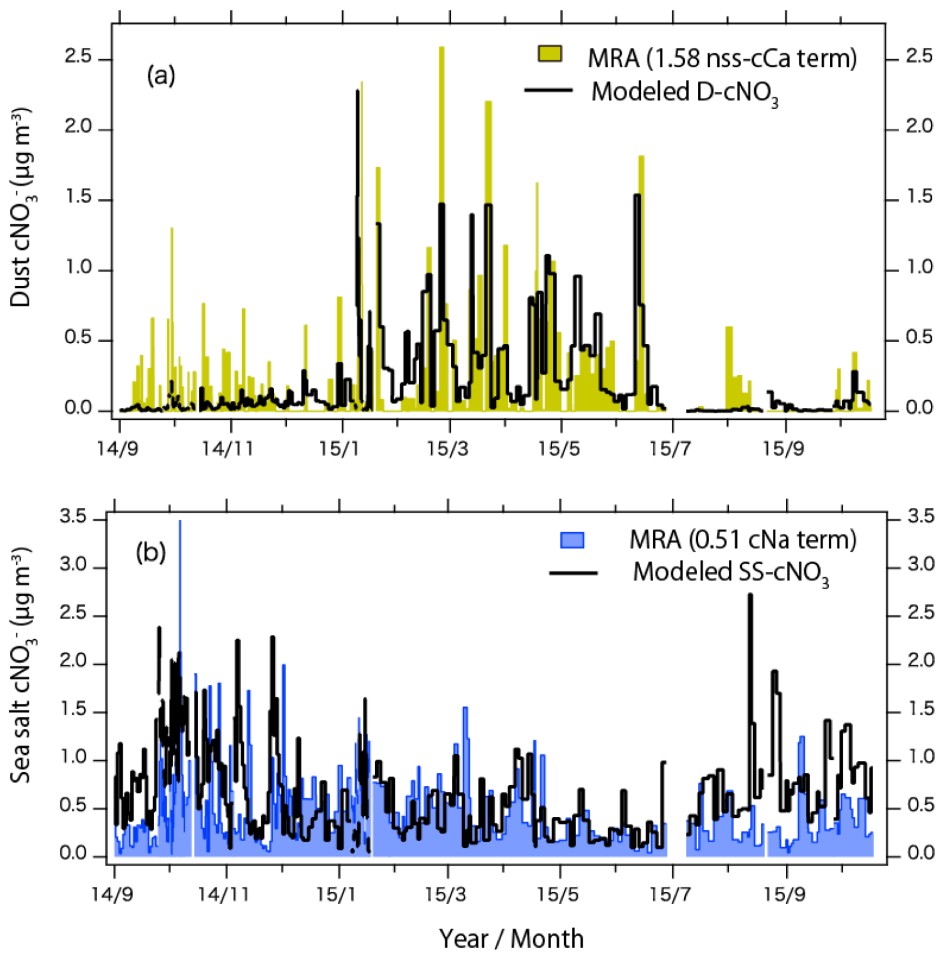

**Figure 9. (a)** D-cNO$_3^-$ estimated by multiple regression analysis and modeled D-cNO$_3^-$, **(b)** the same parameters for sea salt-NO$_3^-$. Modeled results averaged over the same time period as for the D-F measurements.





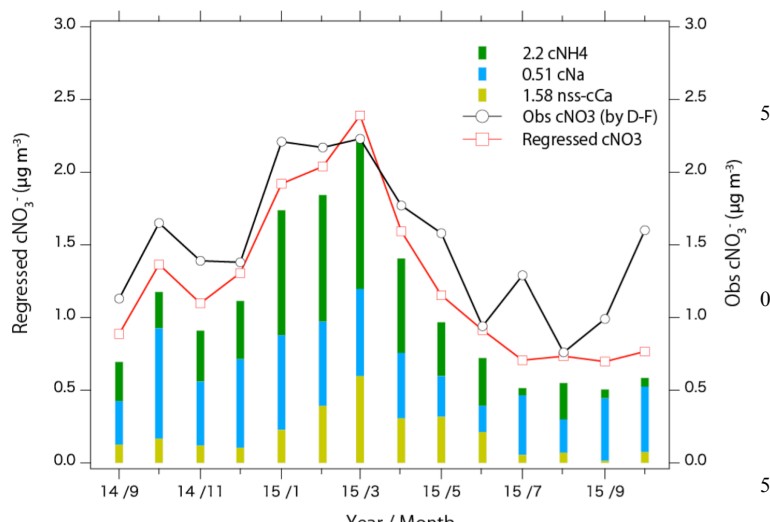

**Figure 10. Monthly average multiple regression results for the dust-cNO$_3^-$, sea salt-cNO$_3^-$, and cNH$_4^+$-related components.**

**Appendix**

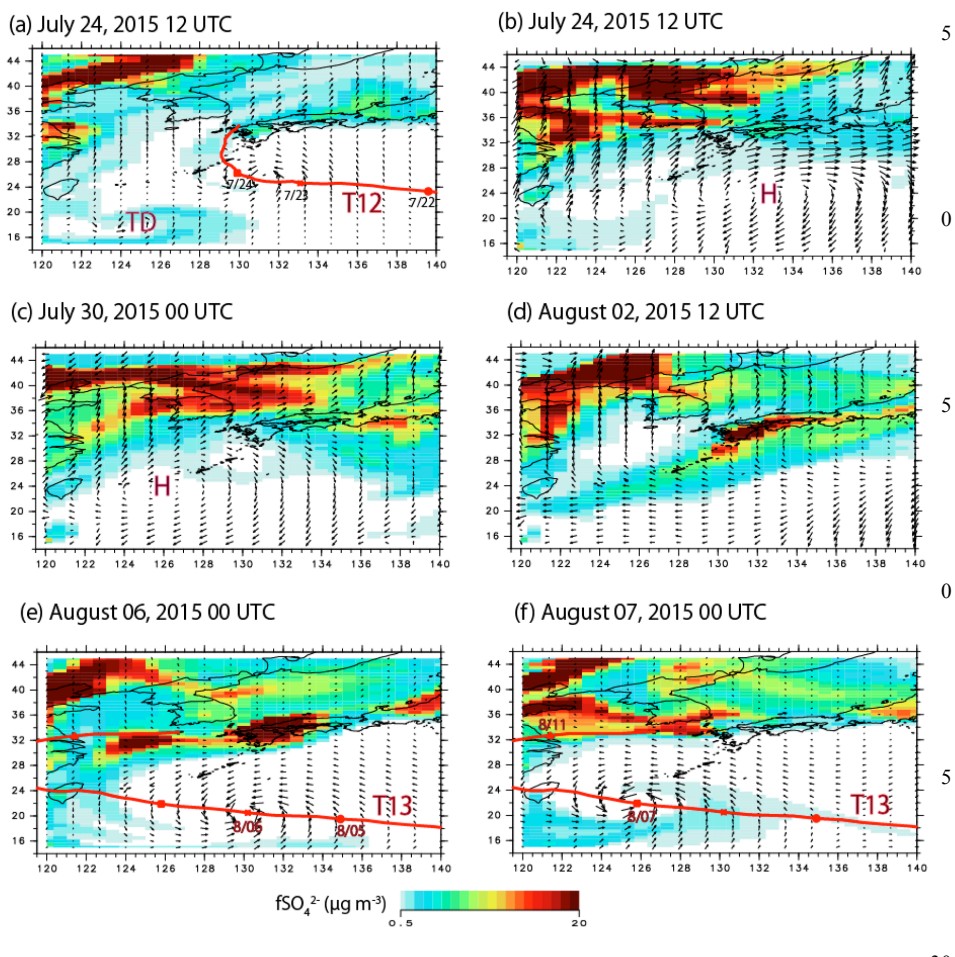

**Figure A1.** Horizontal distribution of fine $SO_4^{2-}$ (color) with wind vectors from model's first vertical level (65 m). The red line indicates the trajectory of Typhoon Halola (T12) in (a), and Typhoon Soudelor (T13) in (e) and (f).





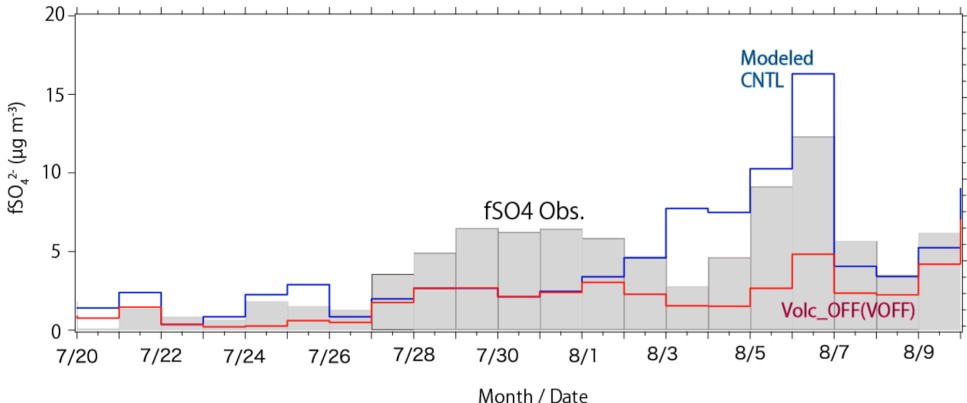

Figure A2. Daily average variation in observed (gray shading) and modeled fine $SO_4^{2-}$ (blue line = CNTL and red line = VOFF).





**Table 1.** Comparison of coarse-mode $NO_3^-$ levels ($\mu g\ m^{-3}$)

| Period | Observation | CTM | | | Mult. Regress. Analysis | | |
|---|---|---|---|---|---|---|---|
| | $cNO_3^-$ | $D\text{-}cNO_3^-$ | $SS\text{-}cNO_3^-$ | $Total\text{-}cNO_3^-$ | $D\text{-}cNO_3^-$ | $SS\text{-}cNO_3^-$ | $cNH_4$ |
| Jan.−June | 1.29[a] (1.82)[b] | 0.40 | 0.43 | 0.83 | 0.34 | 0.46 | 0.68 |
| Sept.−Dec., July | 0.94 (1.37) | 0.06 | 0.73 | 0.79 | 0.11 | 0.50 | 0.26 |
| Annual[c] | 1.13 (1.61) | 0.24 | 0.58 | 0.82 | 0.24 | 0.48 | 0.49 |

a  ACSA observation.

b  D-F observation.

c  Average level between Sept. 2014 and July 2015 (excluding typhoon period).