# Peer review of "Seasonal variation of fine- and coarse-mode nitrates and related aerosols over East Asia: Synergetic observations and chemical transport model analysis"

_Atmospheric Chemistry and Physics, 2017_

## Referee Comment (RC1) · Anonymous Referee #1 · 11 Aug 2017

This paper is well written and focuses on the fine and course aerosols (especially nitrate) transport and transformation processes in long-range transport (LRT) from China to Japan. The authors analysis the transformation process including heterogeneous reaction of SO4–, NO3-, NH4+ and their precursor gases by using the Chemical Transport Model (CTM) and measurement system. The author's CTM model reproduces well the temporal variation of aerosol observation in Kyushu. The authors demonstrate that the course particle is majority of total nitrate and the heterogeneous formation of dust and sea-salt nitrates is important process of course nitrate. Additionally, the authors

suggest a critical importance of inclusion of aerosol microphysical processes in nitrate modeling. This paper is a leading study in which the formation mechanism of nitrate in East Asia is analyzed quantitatively by using the CTM and aerosol observation.

Minor comments: 1. Line 5 of page 3: It is better that "We also examined" is changed to "We focused on" because the nitrate analysis is strong point in this paper.

2. Chapter 2: Monitoring site information should be added.

3. Line 11 of page 3: "Aerosol Chemical Speciation Analyzer and NHx measurement" is better.

4. Lines 18-19 of page 5: Which version of EDGAR is used in this paper? The targeted year of EDGAR and REAS? Which emission inventory for volcanic SO2 is used? Biomass burning emission is included in the simulation?

5. Lines 14-15 of page 6: "The precipitation difference is important for ***" is not clear. Some explanation needs to be added.

6. Line 12 of page 7: Why the high CO is a product of LRT? High CO concentration may be influenced by local emission sources.

7. Line 14 of page 8: "The increase in" should be deleted?

8. Line 23 of page 9: It is better that "(a) NHx" is changed to "(a) f NH4+, C NH4+, and NH3" such as "(b)".

9. Line 1 of page 10: Why is the modeled HNO3 overestimated?

10. Figure 1: The time should be considered in the unit of SO2 emissions such as "Kg/year/grid".

11. Figures 3, 5, 6, 8, 10 and A1: The number of "0" or "5" outside right axis should be deleted.
* * *
[Figure]

2017.

---

## Referee Comment (RC2) · Anonymous Referee #2 · 16 Aug 2017

The authors investigated inorganic particulate nitrate, ammonium, sulfate, and BC over an observation station at Fukuoka in East Asia using data from a 14-month observation and the corresponding GEOS-Chem model simulation. They analyzed the data over daily to seasonal variations and in fine and coarse mode particles, particularly for particulate nitrate. They also include the precursor gas tracers of $NH_3$ and $HNO_3$ in the discussion of their aerosol counterparts $NH_4^+$ and $NO_3^-$, respectively. This is an interesting and valuable study. However, the analysis could be more logical and concise. The scientific accuracy of its statistical analysis is problematic. A major revision

is required before the paper is published in ACP.

Major Comments It would be better to re-organize the discussion content and clearly point out the unique information shown by each figure. I feel it is a bit difficult to follow the discussion that jumps back forth among figures. For example, section 4.3.4 (monthly variation in total NO3 and NH3) starts with a discussion on Figure 8 and follows up with Figure 5d, Figure 6a, and Figure 4a, c, d. I suggest merging Figure 3 and 4 together for daily variation and Figure 5 and 6 together for monthly variation. Please use monthly data to give seasonal and statistical findings and daily data to add any new findings on a daily time scale.

Another major concern is the scientific accuracy of the method used by the authors in explaining the relationship between model and observation based on the regression equation. For example, the authors concluded that the modeled PM2.5 value is approximately 58% of the observed value because the data regression gives PM2.5(model) = 0.58PM2.5(observation)+1.16. From my understanding, a slope of 0.58 does not necessarily lead to that conclusion. A slope that is less than 1 could be a natural phenomenon owing to the representation issue of the model and observation. The GEOS-Chem model data is an average value over a grid box of $0.5°$ x $0.667°$, while the observation is conducted over a station. Consequently, it is hard for the model to capture the maximum and minimum observational values. In other words, the simulated model results tend to be underestimated at the high end and overestimated at the low end, which results in a slope less than 1. The authors need to modify their explanation of the model-observation relationship throughout the manuscript.

Specific Comments 1. Abstract, lines 19-20 and lines 24-26 are duplicates. 2. Page 4 line 19: What are the secondary inorganic aerosols considered for this study? 3. Page 5 line 27: Do NH3 emissions from agriculture and domestic animals belong to the emission of "anthropogenic emissions" mentioned here? 4. Page 6 line 10: How does GEOS-Chem convert coarse-mode sea salt to Na+ concentration? 5. Page 6 line 23 to Page 10 line 15: Reorganize the analysis as "Daily variation in PM2.5,

[Figure]

PM10, aerosol compositions and CO" and "Monthly variation in PM2.5 and aerosol composition". 6. Page 6 line 24: Please explain optical BC. What is its relationship with BC mass concentration? 7. Page 6 line 25: What does "(2h)" mean here? 8. Page 7 lines 9-14: Why do the authors show CO data in this paper? Is there any way to combine the analyses of CO and aerosols? 9. Page 7 line 12: Why does is the high CO a product of long-range transport? 10. Page 7 lines 21-25: Merge Figure 5 and 6 since they are both for monthly average data and NH4 related to both SO4 and NO3. 11. Page 8 line 1: What does "intermittent high" mean here? 12. Page 8 lines 3-4: What is the SF1? Why do the authors mention SF1 event specifically here? Figure 4a shows that the model completely misses the observed peak SO4. 13. Page 9 lines 3-4: Why is cNO3 higher during fall? 14. Page 9 line 28: In sentence ". . . and NH3 gas is also higher even in winter", "higher" than what? 15. Page 9 lines 29-31: How do T and precipitation influence NH3? 16. Page 10 line 4: Using "decreased bias" here may be misleading. My understanding is that the missing emission in model results in an increased emission bias. 17. Page 10 lines 4-9: How do you know that the HNO3-high (-low) and NH3-low (-high) relationship may retain the same equilibrium constant? Is there any evidence to support the authors' conclusion? There are series of aqueous reactions between gases (i.e. NH3 and HNO3) and solid aerosol (i.e. NH4NO3). In NH3 poor regime, NH3 emission cloud matters. 18. Figure 4b and c: Which of the two observation data shown in the figure does "Obs" in the regression equation refer to?

Technique Corrections 1. Abstract lines 21-24: Please change this to "Observational data confirmed that coarse NO3 (cNO3) made up the largest proportion (i.e. 40-55%) of total nitrate (defined as the sum of fNO3, cNO3, and HNO3) during the winter, while HNO3 gas constituted appaoximately 40% of total nitrate in summer. fNO3 peaked during the winter." 2. Abstract line 24: Change "A" to "The". 3. Page 4 line 14: Delete "the" before "a" 4. Page 10 line 1: add "(Fig. 5d)" after "NH3" and add "(Fig. 6a)" after "HNO3". 5. Page 10 line 2: There is no figure Fig. 4d.

---

## Author Comment (AC1) · 3 Oct 2017

This paper is well written and focuses on the fine and course aerosols (especially nitrate) transport and transformation processes in long-range transport (LRT) from China to Japan. The authors analysis the transformation process including heterogeneous reaction of $SO_4-$, $NO_3-$, $NH_4+$ and their precursor gases by using the Chemical Trans- port Model (CTM) and measurement system. The author's CTM model reproduces well the temporal variation of aerosol observation in Kyushu. The authors demonstrate that the coarse particle is majority of total nitrate and the heterogeneous formation of dust and sea-salt nitrates is important process of course nitrate. Additionally, the authors suggest a critical importance of inclusion of aerosol microphysical processes in nitrate modeling. This paper is a leading study in which the formation mechanism of nitrate in East Asia is analyzed quantitatively by using the CTM and aerosol observation.

Thank you very much for your kind reviewing of our manuscript. We have revised our paper based on two reviewer comments, and provide a point-by – point response as below. The revisions are indicated in red in our revised manuscript. We hope our revisions are sufficient for your comments.

**Minor comments:**

1. Line 5 of page 3: It is better that "We also examined" is changed to "We focused on" because the nitrate analysis is strong point in this paper.
   Reply: Thanks. We corrected.

2. Chapter 2: Monitoring site information should be added.
   Reply: We added basic information around the monitoring site.

3. Line 11 of page 3: "Aerosol Chemical Speciation Analyzer and NHx measurement" is better.
   Reply: Thanks. We corrected.

4.  Lines 18-19 of page 5: Which version of EDGAR is used in this paper? The targeted year of EDGAR and REAS? Which emission inventory for volcanic SO2 is used? Biomass burning emission is included in the simulation?

    Reply: We revised by including of the version of EDGAR (Ver. 3) and REAS (Ver. 2.1). Volcanic $SO_2$ emission is based on the Japan Meteorological Agency's data base (we included the URL). Biomass burning information like GFED is not included in our simulation.

5.  Lines 14-15 of page 6: "The precipitation difference is important for ***" is not clear. Some explanation needs to be added.

    Reply: We added some explanations why the precipitation is important for $NH_3$ emission intensity. Please see reply for comment of reviewer 2–15.

6.  Line 12 of page 7: Why the high CO is a product of LRT? High CO concentration may be influenced by local emission sources.

    Reply: Reviewer 2 also commented for CO comparison. Peak concentration of CO and $SO_4$ usually observed simultaneously, so we believe that high CO peak may be influenced by Chinese CO emission. However, the inclusion of CO result is not critically important for our purpose, so we removed the discussion for CO from revised manuscript.

7.  Line 14 of page 8: "The increase in" should be deleted?

    Reply: Thanks. We corrected.

8.  Line 23 of page 9: It is better that "(a) NHx" is changed to "(a) f NH4+, C NH4+, and NH3" such as "(b)".

    Reply: Thanks. We revised.

9.  Line 1 of page 10: Why is the modeled HNO3 overestimated?

    Reply:  Because the $NH_4NO_3$ equilibrium between $NH_3$ and $HNO_3$ is

given as

$$HNO_{3\ (g)} + NH_{3\ (g)} \rightleftarrows NH_4NO_{3\ (p)}$$

In equilibrium condition, if $NH_3$ is small, then $HNO_3$ could be higher to keep equilibrium.

10. Figure 1: The time should be considered in the unit of SO2 emissions such as "Kg/year/grid".
    Reply: Thanks. We corrected.

11. Figures 3, 5, 6, 8, 10 and A1: The number of "0" or "5" outside right axis should be deleted.
    Reply: We removed the line numbers from the figure pages.

---

## Author Comment (AC2) · 3 Oct 2017

The authors investigated inorganic particulate nitrate, ammonium, sulfate, and BC over an observation station at Fukuoka in East Asia using data from a 14-month observation and the corresponding GEOS-Chem model simulation. They analyzed the data over daily to seasonal variations and in fine and coarse mode particles, particularly for particulate nitrate. They also include the precursor gas tracers of NH3 and HNO3 in the discussion of their aerosol counterparts NH4+ and NO3-, respectively. This is an interesting and valuable study. However, the analysis could be more logical and concise. The scientific accuracy of its statistical analysis is problematic. A major revision is required before the paper is published in ACP.

Thank you very much for your kind reviewing of our manuscript. We revised our paper based on two reviewer comments and provide a point-by –point response as below. We also removed the discussion of $NH_3$ and $HNO_3$ balancing for $NO_3$ formation, and revised the statistical analysis including the normalized mean bias (new Table1). The revisions are indicated in red in our revised manuscript. We hope our revisions are sufficient for your comments.

**Major Comments**

2-A

It would be better to re-organize the discussion content and clearly point out the unique information shown by each figure. I feel it is a bit difficult to follow the discussion that jumps back forth among figures. For example, section 4.3.4 (monthly variation in total NO3 and NH3) starts with a discussion on Figure 8 and fol- lows up with Figure 5d, Figure 6a, and Figure 4a, c, d. I suggest merging Figure 3 and 4 together for daily variation and Figure 5 and 6 together for monthly variation. Please use monthly data to give seasonal and statistical findings and daily data to add any new findings on a daily time scale.

Reply: We re-organized the section 4. We also merge figures 3 and 4 as new Figure 3, and figures 5 and 6 as new Figure 4. We re-organized the discussion section in order to clarify the monthly (seasonal) variation of aerosols (this is our final purpose). However, the monthly mean value depends on the how much model can reproduce the intermittent (i.e., 1 − 2 times/ week) long-range transport from Asian continent to Japan, and sporadic dust transport. To show these intermittent and sporadic phenomena, we used the daily average concentration change (new Figure 3), and calculate the model statistics (add new Table 1 to show the model performance).

2-B

Another major concern is the scientific accuracy of the method used by the authors in explaining the relationship between model and observation based on the regression equation. For example, the authors concluded that the modeled PM2.5 value is approximately 58% of the observed value because the data regression gives PM2.5(model) = 0.58PM2.5(observation)+1.16. From my understanding, a slope of 0.58 does not necessarily lead to that conclusion. A slope that is less than 1 could be a natural phenomenon owing to the representation issue of the model and observation. The GEOS-Chem model data is an average value over a grid box of 0.5∘ x 0.667∘, while the observation is conducted over a station. Consequently, it is hard for the model to capture the maximum and minimum observational values. In other words, the simulated model results tend to be underestimated at the high end and overestimated at the low end, which results in a slope less than 1. The authors need to modify their explanation of the model-observation relationship throughout the manuscript.

Reply: We revised the discussion mainly base on the Normalized Mean Bias (NMB) and other model statistics as shown in new Table 1. We also include the model representation issue into the text.

**Specific Comments**

2-1.   Abstract, lines 19-20 and lines 24-26 are duplicates.

Reply: Thanks. We corrected.

2-2.   Page 4 line 19: What are the secondary inorganic aerosols considered for this study?

Reply: We included the component of secondary inorganic aerosols ($SO_4$, $NO_3$, $NH_4$)。 As describe in text, we do not count secondary OC.

2-3.   Page 5 line 27: Do NH3 emissions from agriculture and domestic animals belong to the emission of "anthropogenic emissions" mentioned here?

Reply: REAS $NH_3$ inventory includes fertilizer application and livestock excreta. In our model sensitivity study, we reduce (20% reduction) all the emission intensity including NH3. To remove the confusion, we reword "all emissions" instead of "anthropogenic emissions".

2-4.   Page 6 line 10: How does GEOS-Chem convert coarse-mode sea salt to Na+ concentration?

Reply: GEOS-Chem simulated coarse mode sea-salt mass concentration was converted to Na mass based on the salinity and Na mass ratio of sea water (Keene et al., 1986).

Keene. W.C. A.A.P. Pszenny, J.N. Galloway, M.E. Hawley: Sea-salt corrections and interpretation of constituent ratios in marine precipitation, J. Geophysical Research, 91, 6646-6658, 1986.

2-5.   Page 6 line 23 to Page 10 line 15: Reorganize the analysis as "Daily variation in PM2.5, PM10, aerosol compositions and CO" and "Monthly variation in PM2.5 and aerosol composition".

Reply: Please see the reply for 2-A. We clearly separated the

discussion for daily (for mainly model validation) and monthly mean variation.

2-6.   Page 6 line 24: Please explain optical BC. What is its relationship with BC mass concentration?
Reply: Optical BC was measured by NIR light scattering method, and observed data had a good correlation with the IMPROVE protocol measurement (Hasegawa et al., 2004). We add this text in Section 2.1.

Hasegawa, S., S. Wakamatsu, K. Tanabe: Parallel measurement test of black carbon monitors, Proceeding of 21th Symposium on Aerosol Science and Technology, Japan Association of Aerosol Science and Technology, p. 7, 2004.

2-7.   Page 6 line 25: What does "(2h)" mean here?
Reply: Sorry. This is our mistake. All the plot in Figures 3 and 4 are daily average. We corrected the manuscript.

2-8.   Page 7 lines 9-14: Why do the authors show CO data in this paper? Is there any way to combine the analyses of CO and aerosols?
Reply: Please also see the reply for 1-6. We removed the discussion of CO from our revision.

2-9.   Page 7 line 12: Why does is the high CO a product of long-range transport?
Reply: We removed this discussion.

2-10.  Page 7 lines 21-25: Merge Figure 5 and 6 since they are both for monthly average data and NH4 related to both SO4 and NO3.
Reply: Thanks We merged Figure 5 & 6 and make new Figure 4.

2-11.  Page 8 line 1: What does "intermittent high" mean here?

Reply: Long-range transport of $SO_4$ from Asian continent to Japan occurs once or twice / week during winter to early spring (It depends on synoptic weather changes). This occurs intermittently. We added a short description of this phenomena and reference.

2-12. Page 8 lines 3-4: What is the SF1? Why do the authors mention SF1 event specifically here? Figure 4a shows that the model completely misses the observed peak SO4.

Reply: SF1 event was used to identify the period of volcano impact is important for $SO_4$ level (please see Figures A1 and A2). In order to identify this meaning, we changed to" SVolc" in our revision. Because the volcanic emission is highly natural phenomena and hard to predict the day by day changes, so the modeled $SO_4$ level during SVolc misses the observed peak $SO_4$ (indicating the modeled $SO_2$ emission is under-estimated). We added some detailed explanation into the text and appendix.

2-13. Page 9 lines 3-4: Why is cNO3 higher during fall?

Reply: This is due to the formation of sea-salt nitrate. We revised the text in order to explain this point.

2-14. Page 9 line 28: In sentence ". . . and NH3 gas is also higher even in winter", "higher" than what?

Reply: Sorry for confusion. We revised this part as "$NH_3$ gas level exceeded 1 µg/m$^3$ (i.e., 0.4 ppb) even in winter".

2-15. Page 9 lines 29-31: How do T and precipitation influence NH3?

Reply: We revised our text as follows: The high $NH_3$ concentration (see Fig. 5d) in August 2015 (3 – 4 times higher than August 2014) might be due to differences in high temperature (monthly mean was 28.4°C, which is 1.7°C higher than 2014) and less precipitation (month total was 186mm, which is 228mm smaller than 2014). Roelle *et al.* (2002) indicated that the $NH_3$ emission from soil increased exponentially as

soil temperature increase, and more soil water due to precipitation filled the pores in the soil matrix and hinder the diffusion of $NH_3$ from the soil to the air. Their results suggested that our observed meteorological conditions in each year can be the reasons for NH3 concentration variation.

Roelle, P. A., Aneja, V. P.: Characterization of Ammonia Emissions from Soils in the Upper Coastal Plain, North Carolina. *Atmos. Environ.*, **36**, 1087-1097 (2002)

2-16.  Page 10 line 4: Using "decreased bias" here may be misleading. My understanding is that the missing emission in model results in an increased emission bias.

Reply: Thanks. We corrected.

2-17.  Page 10 lines 4-9: How do you know that the HNO3-high (-low) and NH3-low (-high) relationship may retain the same equilibrium constant? Is there any evidence to support the authors' conclusion? There are series of aqueous reactions between gases (i.e. NH3 and HNO3) and solid aerosol (i.e. NH4NO3). In NH3 poor regime, NH3 emission cloud matters.

Reply: Thank you very much for your comment. We agreed this point, and removed this part of discussion from the text.

2-18.  Figure 4b and c: Which of the two observation data shown in the figure does "Obs" in the regression equation refer to?

Reply: Daily averaged value from ACSA data was used to "Obs". We identify this point into figure.

**Technique Corrections**

2-T1.  Abstract lines 21-24: Please change this to "Observational data confirmed that coarse NO3 (cNO3) made up the largest proportion (i.e.

40-55%) of total nitrate (defined as the sum of fNO3, cNO3, and HNO3) during the winter, while HNO3 gas constituted appaoximately 40% of total nitrate in summer. fNO3 peaked during the winter."

2-T2.  Abstract line 24: Change "A" to "The".

2-T3.  Page 4 line 14: Delete "the" before "a"

2-T4.  Page 10 line 1: add "(Fig. 5d)" after "NH3" and add "(Fig. 6a)" after "HNO3".

2-T5.  Page 10 line 2: There is no figure Fig. 4d.

Reply: For all technique corrections, we corrected. Thank you very much for your kind helps.